# Next Generation Sequencing in Non-Small Cell Lung Cancer: Pitfalls and Opportunities

**DOI:** 10.3390/diagnostics10121092

**Published:** 2020-12-15

**Authors:** Chiara Lazzari, Alessandra Bulotta, Maria Giulia Cangi, Gabriele Bucci, Lorenza Pecciarini, Silvia Bonfiglio, Vincenza Lorusso, Stefania Ippati, Gianluigi Arrigoni, Greta Grassini, Claudio Doglioni, Vanesa Gregorc

**Affiliations:** 1Department of Oncology, IRCCS San Raffaele, 20132 Milan, Italy; bulotta.alessandra@hsr.it (A.B.); lorusso.vincenza@hsr.it (V.L.); ippati.stefania@hsr.it (S.I.); gregorc.vanesa@hsr.it (V.G.); 2Department of Pathology, IRCCS San Raffaele, 20132 Milan, Italy; cangi.mariagiulia@hsr.it (M.G.C.); pecciarini.lorenza@hsr.it (L.P.); arrigoni.gianluigi@hsr.it (G.A.); grassini.greta@hsr.it (G.G.); doglioni.claudio@hsr.it (C.D.); 3Center for Omics Science, IRCCS San Raffaele, 20132 Milan, Italy; bucci.gabriele@hsr.it (G.B.); bonfiglio.silvia@hsr.it (S.B.)

**Keywords:** non-small cell lung cancer, next-generation sequencing, molecular tumor board

## Abstract

Lung cancer remains the first cause of cancer-related deaths worldwide. Thanks to the improvement in the knowledge of the biology of non-small cell lung cancer (NSCLC), patients’ survival has significantly improved. A growing number of targetable molecular alterations have been identified. Next-generation sequencing (NGS) has become one of the methodologies entered in clinical practice and was recently recommended by the European society for medical oncology (ESMO) to perform a comprehensive molecular characterization in patients with cancer. The current review provides an overview of the clinical trials that have explored the impact of NGS in patients with cancer, its limits, and advantages.

## 1. Introduction

Lung cancer remains the first cause of cancer-related deaths worldwide [1]. Non-small cell lung cancer (NSCLC) accounts for approximately 80–85% of cases. The majority of patients are diagnosed with advanced or metastatic disease.

Thanks to technological advances, the molecular landscape of NSCLC has evolved, with the emergence of rare but clinically actionable subtypes of NSCLC tumors [2]. Currently, molecular classification has become an essential part of routine cancer care. The efforts to molecularly classify patients and tailor treatment to their biological profile have significantly prolonged patients’ survival [3]. During the last 10 years, a growing number of targetable molecular alterations have been identified. New compounds have entered the therapeutic portfolio, thus further improving the complexity of NSCLC tumor biology and the treatment opportunities for patients.

Despite this progress and the importance of molecular tumor profiling, the molecular assessment remains heterogeneous, mainly due to institutional organization issues. Nationwide programs have shown the feasibility of large-scale molecular screening [3,4], demonstrating that consortia between different institutions represent a successful strategy to increase the number of patients with a molecular classification.

The complexity of tumor genotyping has favored the development of cost-effective technologies, sustainable for the healthcare system, and able to provide reproducible information of clinical value. In this scenario, next-generation sequencing (NGS) has become one of the methodologies entered in clinical practice and recently recommended by the European society for medical oncology (ESMO) to obtain comprehensive molecular profiling in patients with metastatic lung adenocarcinoma, prostate cancer, ovarian cancer, or cholangiocarcinoma (https://www.esmo.org/newsroom/press-office/esmo-issues-first-recommendations-on-using-next-generation-sequencing-for-advanced-cancers). However, the real impact of NGS on the routine clinical practice remains to be further determined. The current review provides an overview of the critical issues, the advantages, and the limits of using NGS for the biological characterization of patients with NSCLC. We will examine the technical requirements for efficient NGS profiling and the clinical trials that have explored its impact on the management of patients with NSCLC.

## 2. Next-Generation Sequencing: Tissue Requirements

Conversely, from single-gene sequencing, NGS has the potential to concurrently detect single nucleotide variants, small insertions, deletions, copy number alterations, structural variants, gene fusions, or chromosomal rearrangements [5], thus reducing the costs and sparing the time for a comprehensive molecular assessment (Figure 1). NGS allows to identify co-occurring genetic alterations in a specific subgroup of molecularly classified patients and better define the genomic complexity and the prognosis of patients with cancer, as in the case of KRAS mutant [6] or EGFR mutant NSCLC tumors [7,8]. The assay’s reliability and sensitivity constitute essential requirements since NGS should be non-inferior to standard diagnostics and to generate clinically useful information.

The whole process (from tissue collection to the molecular report) needs to be structured to guarantee the quality of the results. One of the significant issues in the molecular profiling of patients with advanced NSCLC is related to the amount of tumor tissue collected at the time of diagnosis and required for comprehensive molecular characterization. In approximately 23% of patients [9], the tissue is not adequate for molecular analyses due to the difficulty of reaching tumor sites with non-invasive methods or patients’ comorbidities. Moreover, in some cases, tumor necrotic cells decrease cellular density, resulting in low-quality sequencing.

The limit of detection of NGS is dependent on the percentage of available tumor cells. Currently, there is not a validated cut-off of tumor cells required for NGS analysis [10]. Furthermore, due to the small amount of tissue derived from biopsy, in approximately 30% of patients diagnosed with NSCLC, tumor cellularity is <40%. Data from literature suggest that the minimum tumor cell content recommended should be more than two times the limit of detection of the test used [10]. Based on previous experiences, NGS can detect single nucleotide variants in samples with tumor cellularity >20%, while the probability of identifying molecular alterations in samples with tumor cellularity between 10–20% decreases [11]. To detect and not miss potentially targetable gene alterations, the pathology assessment for sample adequacy represents one of the essential requirements before proceeding with genomic profiling (Figure 2). Sample collection, sample selection, sample processing, and quality review need to become standardized procedures. The cellular density, the presence of necrosis or fibrosis, and the percentage of tumor cells represent the principal features to define the tumor samples’ adequacy.

## 3. Next-Generation Sequencing: Bioinformatic Issues

The introduction of NGS in clinical practice has been accompanied by issues of ethical, administrative, technological, scientific, financial, and organizational nature [12]. The interpretation of the results, the bioinformatic workflow, the storage of the data with the respect of the General Data Protection Regulation (GDPR) legislation, and the building of the genetic report, including the relevant molecular results with prognostic or predictive value, helpful for therapeutic decisions, remain open issues.

Due to the large amount of NGS information and the legal consequences derived from the profiling assessment, article 22 of the GDPR discourages defining treatment based on automated processing. To overcome this issue, in parallel with the introduction of NGS, molecular tumor boards have been created (Figure 2). Molecular tumor boards include experts in genetics, oncology, bioinformatics, molecular biology, and pathology. They discuss the NGS findings to interpret, differentiate the molecular alterations identified into the pathogenic driver or passenger mutations, and translate molecular data into treatment options. Rank the molecular variants observed according to the level of evidence for actionability, interpret their functional relevance, evaluate the associated cancer-related pathways, analyze the alterations of unknown or uncertain significance, define whether the aberrations identified in different histologies have a comparable biologic function, and search for the availability of clinical trials represent the strategy to improve the therapeutic opportunities for patients with cancer. Within the molecular tumor boards, discussions should be based on the critical evaluation of the literature and databases’ comparison. All the somatic, nonsynonymous variants are generally compared with a precision oncology database, like the Memorial Sloan Kettering Cancer Center’s OncoKB [13] or the Precision Medicine Knowledge Base (PMKB) from the Weill-Cornell University, which both annotate the biologic, oncogenic relevance, and the prognostic or predictive significance of the molecular alterations identified. Those variants not included in such bases might be further analyzed using other cancer databases, including the Catalogue of Somatic Mutations in Cancer (COSMIC), The Cancer Genome Atlas (TCGA), or International Cancer Genome Consortium (ICGC). Thanks to comparison, along with predicted alleles frequency, it is possible to identify putative driver alterations. Moreover, each patient might harbor from a few to multiple somatic variants. To facilitate their interpretation, based on their clinical significance in cancer diagnosis, prognosis, and/or therapeutics, the alterations identified are classified into ‘tiers’. Different classification systems exist, and no global consensus has been adopted among the different national and local molecular tumor boards. Only recently, both the American AMP/ASCO/CAP and the European ESMO-ESCAT published recommendations and guidelines to rank genomic alteration. The four categories adopted by the American consortium [10] are reported in Table 1, and include: Tier 1 (variants with strong clinical significance);Tier 2 (variants with potential clinical significance);Tier 3 (variants of unknown clinical significance)Tier 4 (variants deemed benign or likely benign).

The European system [14] suggested a slightly different tiering system, based on six levels of importance:Tier I (targets ready for implementation in routine clinical decisions);Tier II (investigational targets that likely define a patient population that benefits from a targeted drug, but additional data are needed);Tier III (clinical benefit previously demonstrated in other tumor types or for similar molecular targets);Tier IV (preclinical evidence of actionability)Tier V (evidence supporting co-targeting approaches; and tier X, lack of evidence for actionability).

Beyond the complexity of NGS interpretation, the ethical management of NGS’s information, especially in those cases in which germline variants are included or when a limited number of approved drugs are available, represent further challenges in the era of precision oncology. To overcome these issues, genetic counseling is recommended, and the design of biomarker-driven clinical trials is expected together with the development of the NGS assays.

## 4. Previous and Ongoing Experiences with NGS in Patients with NSCLC

Several prospective phase II nationwide protocols have been developed to test the clinical utility of precision medicine in treating patients with cancer. They have been designed as “umbrella” or ‘basket’ trials. The “umbrella” design considers multiple genomic alterations within the same cancer histology, while the “basket” design analyses the same genomic alteration across different tumor types (Table 2).

The SHIVA was the first randomized, multicenter, prospective, phase II study, designed to compare in patients with advanced solid tumors, previously progressing to standard therapy, the progression-free survival (PFS) improvement of targeted agents, selected according to patients’ tumor molecular profiling, over standard treatment [15]. Matched therapy included drugs approved for clinical use but outside their indications. A pre-defined algorithm was used for treatment allocation, and 46 genes were tested. Among the patients enrolled (741), only in 67% of the cases (496) the molecular characterization was feasible, and in 40% of them (293), a molecular alteration matching with one targeting agent was identified. No significant PFS improvement was observed in patients receiving matched targeted agents. However, for the first time, the feasibility of conducting a randomized precision medicine trial was proved. There are different reasons for the failure of the SHIVA study, including the treatment algorithm applied to define and prioritize the molecular driver alterations identified, and the targeted therapies used, some of which lack activity.

The MOSCATO-01 was a monocentric, prospective trial enrolling, between December 2011 and March 2016, 1035 patients with unresectable or metastatic solid tumors, previously progressing during at least one line of therapy [16]. There were 1035 enrolled, and in 82% of the cases, tissue was adequate for molecular analysis, and 170 had a diagnosis of lung cancer. For biopsies collected from May to November 2012, the targeted gene panel covered 190 amplicons in 40 genes. For biopsies collected from December 2012 to September 2013, 207 amplicons in 50 genes were analyzed, while for biopsies collected after September 2013, the targeted sequencing panel included the analysis of 75 oncogenes or tumor suppressors. An actionable target was identified in 411 patients. Among these, 199 received the matched therapy, while treatment was not started in the other patients because of clinical deterioration, the absence of tumor progression, the lack of available slots, or the patients’ refusal. The trial reached its primary objective, demonstrating a longer PFS under matched treatment, compared to the PFS observed during the previous therapy.

The BATTLE-2 was a phase II randomized study designed to evaluate the efficacy of targeted therapies in KRAS mutated NSCLC, refractory to platinum-based chemotherapy [17]. Advanced patients with NSCLC not harboring EGFR activating mutations or EML4-ALK rearrangement were enrolled, and the tissue was analyzed by gene expression profile and by the Foundation Medicine NGS sequencing panel. There were 54 KRAS mutated patients included, and randomized between erlotinib, erlotinib plus the AKT inhibitor MK-2206, the MEK inhibitor AZD6244 plus the AKT inhibitor MK-2206, or sorafenib. Results did not demonstrate the benefit of any of the treatment arms. The use of erlotinib was not effective in inhibiting KRAS mutated NSCLC tumors, and the heterogeneous biology of KRAS mutant NSCLC tumors [6] explains the failure of the BATTLE-2 study. Thanks to the use of NGS, co-occurring genetic alterations in genes other than KRAS, responsible for cancer cell proliferation, the development of escape mechanisms, and that impact on patients’ prognosis have been identified. Currently, KRAS mutated NSCLC patients are classified into three clusters: One, including co-mutations in TP53 (KP), a second, carrying inactivating mutations in the tumor suppressor liver kinase b1 (LKB1) (KL), and a third, harboring bi-allelic deletions of two tumor suppressor genes, CDKN2A and CDKN2B (KC) [6]. Those with TP53 mutations have higher overall mutational load compared to the other subtypes and a high expression of genes involved in anti-tumor immunity and the JAK STAT pathway, those included in the KL cluster show the lack of immune system engagement, and those comprised in the KC cluster a low expression of thyroid transcription factor 1 (TTF1), but high expression of markers of mucinous differentiation. Higher activation of MEK/ERK downstream signaling was observed in mice with KRAS mutation only and in those with concurrent p53 mutation, compared with those carrying the LKB1 mutation, where a higher activation of AKT and SRC was found. These data suggest the importance of defining the molecular bases for cancer progression and to identify treatment selected to inhibit specific signaling pathways.

What we have learned from the negative results of the SHIVA and the BATTLE-2 studies is that in order to maximize the clinical benefit of precision medicine in oncology, the establishment of whether tumor cells are dependent or independent for tumor growth from a specific oncogenic pathway is of crucial importance, as the selection of the drugs able to inhibit the function of a driver alteration.

Different trials are currently ongoing. The TAPUR study (NCT02693535) is a phase II, open-label study that aims to evaluate the anti-tumor activity of commercially available targeted anticancer drugs used outside of their FDA-approved indications in patients with advanced solid tumors, non-Hodgkin lymphoma, or multiple myeloma, and no longer benefiting from standard treatment [18]. Participants are enrolled in cohorts defined by tumor type, genomic alteration, and study drug. The primary endpoint within each cohort is objective response rate (ORR) or stable disease of at least 16 weeks. Only in case of a successful outcome for two or more patients, the cohort is expanded to enroll an additional 18 patients. If at least 7 out of the 28 patients enrolled for each cohort have a tumor response, a drug activity signal is declared. Results from two cohorts enrolling patients with NSCLC have been reported recently. One evaluated the efficacy of palbociclib in patients harboring cyclin-dependent kinase Inhibitor 2A (CDKN2A) loss or mutation [19]. Results showed a disease control rate of 29%, a median PFS of 7.9 weeks, and a median OS of 20.6 weeks. In the other cohort, cetuximab was administered in patients without KRAS, NRAS, or v-raf murine sarcoma viral oncogene homolog B1 (BRAF) mutations [20]. No significant clinical activity was observed.

The S1400 Lung-MAP (NCT02154490) is a prospective phase II study, with an “umbrella” adaptive design, enrolling patients with squamous lung tumor (SCC), progressing to first-line therapy [21]. The trial was opened in June 2014 and closed in January 2019, when a new Lung-MAP screening protocol was expanded to include patients with all advanced NSCLC. The aim was to identify and test new targeted treatments. There were 11 sub-studies opened. Of these, 8 have been closed, and 3 are still enrolling patients. Of this, 1064 with SCC have been included, and tissue analyzed by Foundation Medicine NGS panel. Successful biomarker analysis was performed in 91% of the cases. The TP53 gene was the most frequently altered gene observed and was identified in 90% of the patients evaluated. Patients carrying a molecular alteration received a targeted agent. Those in which no targetable genes were identified were randomized between nivolumab + ipilimumab and nivolumab. Despite these huge efforts, few actionable or druggable mutations have been detected in patients with SCC [22,23,24]. However, new targets have emerged, including hotspot mutations in oxidative stress pathway transcription factor, nuclear factor erythroid-derived 2-like 2 (NFE2-L2), and its negative regulator kelch-like ECH-associated protein 1 (KEAP1), and new compounds have been tested with promising preliminary activity. The “umbrella” design has allowed screening of new drugs, and evaluating their preliminary efficacy, thus sparing costs and reducing time.

The National Lung Matrix (NCT02664935) is a phase II trial, designed in advanced NSCLC progressing to standard treatment or who refuse any standard of care in the first-line setting [25]. The NGS panel used analyzes 28 genes and stratifies patients into 22 cohorts according to the molecular alterations identified. Eight drugs are tested. The primary objective is to evaluate whether there is a signal of drug activity. Objective response (OR), durable clinical benefit (DCB, defined as progression-free survival at 24 weeks), and PFS represent the endpoints measured to determine the drug’s clinical activity. Thirty patients are enrolled in each cohort, with futility analyses at 15. Preliminary results from 19 cohorts have been recently published [26]. There were 5467 patients screened, and at least a molecular alteration classified as tier 1 or tier 2 had been identified in 3181 cases. Among these, 2007 patients were eligible for the trial. To the different cohorts, 302 patients were allocated and 289 started treatment, with 1443 patients not included. Among these, 14% had ongoing toxicity, poor performance status, or symptomatic brain metastases, 25% were still receiving first-line treatment, 27% died on previous therapy. Based on the preliminary findings, a limited number of combinations demonstrated a clinically relevant benefit. Currently, eight cohorts remain open and continue with enrollment.

The SAFIR02_Lung (NCT02117167) is a phase II study, enrolling advanced NSCLC patients, wild type for EGFR activating mutations, or EML4-ALK rearrangement, who have stable disease or partial response following four cycles of an induction platinum-based chemotherapy. The trial aims to compare a targeted treatment with standard maintenance therapy (pemetrexed in non-squamous patients and erlotinib in squamous) or immunotherapy with standard maintenance therapy in those patients without actionable genomic alterations.

The NCI-MATCH (NCT02465060) is a phase II trial, enrolling patients with advanced solid tumors, lymphomas, or multiple myeloma who have progressed on standard treatment. The tissue is analyzed by an NGS panel evaluating molecular alterations in 143 genes. The trial was opened in August 2015 and had nearly 40 treatment arms.

The ACC lung protocol is an ongoing Italian prospective study enrolling 1000 advanced naive NSCLC patients from 24 Italian institutes, who are molecularly characterized using a panel designed by academia that analyses 182 genes. The genetic alterations were included if: Associated with predictive value (group 1, “actionable”), of highly probable importance in driving disease progression (group 2, “driver”), associated with altered metabolism of cancer drugs, (group 3, “pharmacogenomic variants”). Genetic alterations were defined “actionable” if they fulfilled one of the following criteria: (i) At least one positive (predicting sensitivity) or negative (predicting resistance) interaction with a specific cancer drug according to the FDA or NCCN guidelines, (ii) prospective trials or retrospective analyses where the alteration was used as a stratification biomarker, or (iii) case reports of exceptional response. The drug-gene interactions were extracted from databases updated as of late 2016 [27,28,29,30]. This analysis yielded 164 genes deemed of potential clinical relevance and 89 “actionable” fusion transcripts, extracted from the Cancer Genome Atlas (TCGA) fusion database relative to NSCLC (http://54.84.12.177/PanCanFusV2/). Genetic alterations were defined as “driver” for NSCLC biology if they were scored by at least 2 out of 6 bioinformatic pipelines for driver identification [31,32,33]. This analysis identified 33 genes. Finally, 141 germline variants from 86 genes, associated with altered pharmacokinetic response to drugs in oncology, extracted from the PharmGKB database (https://www.pharmgkb.org/), were included. The NGS panel was retrospectively tested on archived tumor paraffin and commercial samples. The ongoing prospective trial is evaluating its clinical utility and its diagnostic accuracy in terms of sensitivity and specificity when introduced routinely in the diagnostic work-up of patients with advanced NSCLC. The development of a national database for biomarker tests and the implementation of a national cancer genome screening program represent additional objectives of the trial in order to build a national repository of clinical and genetic information in patients with NSCLC. Patients are treated according to the National Guidelines. Patients carrying EGFR activating mutations receive EGFR-TKIs, those harboring EML4-ALK fusion crizotinib or alectinib, those positive for ROS1 fusion crizotinib, those carrying BRAF V600E dabrafenib + trametinib. Patients with PD-L1 overexpression (≥50%) receive pembrolizumab, while those in which no molecular alteration is observed are treated with standard first-line chemotherapy (platinum-based regimen, chemo-immunotherapy, or mono-chemotherapy) according to the investigator’s choice. In case of molecular alterations in druggable genes different from EGFR, BRAF, EML4-ALK, and ROS1, and for which there are no drugs registered within the National Health System, the availability of opened clinical trials or compassionate use programs is evaluated. In case no clinical trial is available, but in the presence of positive data from at least one phase II study indicating the effectiveness of an approved drug, but outside its indication, a request to AIFA is sent, asking for the reimbursement of the therapy, in agreement with the Article 48 of the Italian Law 2003/326.

## 5. Comments and Conclusions

During the last 15 years, cancer treatment has evolved, and comprehensive molecular characterization has become an essential part of routine cancer care. The appropriate use of genomic profiling has supported precision medicine and has offered new treatment options for patients with cancer.

Standard techniques aimed to hotspot targeted mutation analysis, and used in a sequential highly selective single-gene testing approach, have been part of routine practice for several years. The expansion of available approved targeted drugs has caused an increasing demand for multiplex testing for a more efficient simultaneous detection of targeted alterations. In this context/perspective, NGS allows maximizing the identification of clinically relevant genomic alterations using a limited amount of tissue, thus reducing time and costs. NGS per-sample costs have dramatically dropped over time, and nowadays, targeted NGS analysis for oncologic therapy selection not only has comparable costs to the standard techniques carried on for the analysis of 5–10 different genes but at the same price, it allows wider molecular characterization for more accurate patient stratification. The growing knowledge of the molecular alterations of neoplastic diseases offered by NGS analysis has been changing the diagnostics approach to tumors and also their classification.

The introduction of NGS has been accompanied by ethical and organizational issues. First, to guarantee the quality of NGS results, the whole process (from tissue collection to the molecular report) needs to be structured. Obtaining an adequate amount of tissue remains one of the critical aspects for a reliable result. Moreover, the complexity of NGS findings has required the building of clinical support tools, able to offer detailed, evidence-based information about individual somatic mutations and structural alterations into a standardized and easily interpretable format. Different systems to classify the molecular aberrations are available, and there are not specific guidelines to define how to generate the genomic report. For these reasons, the institution of molecular tumor boards is fundamental to interpret the genetic information and translate into therapeutic options critically. Despite only those molecular alterations classified as “tiers 1” are used in clinical practice to establish the treatment options for patients with cancer, the presence of alterations classified as “tiers 2 or 3” might be useful to define patients’ prognosis. Using a detailed informed consent before NGS testing and communicating NGS results to patients remain key aspects of the process that have raised ethical concerns, and that must always be addressed by the practicing oncologists when ordering a NGS test. Considering the complexity of NGS analysis and the amount of genetic information provided, NGS still remains a methodology to be applied within clinical trials. The design of studies using NGS to molecularly classify patients represents an opportunity to explore new targets and test new compounds, thus evaluating the preliminary efficacy of new drugs, sparing costs, and reducing time.

Though several data analysis over clinical trials showed that molecular diagnosis accounts for only 6% of the cost of molecular-guided therapy per patient and that the costs of drugs and hospitalizations remain the main cost drivers, public health systems should take into account an accurate revision of the whole molecular diagnostics process in order to allocate more adequate resources to Pathology and Oncology Department.

Previous and ongoing experiences have shown that performing a comprehensive molecular characterization is useful to identify not only the driver of the genetic alteration but also co-occurring mutations in specific subgroups of molecularly defined patients. This helps to characterize the signaling pathways of primary or acquired resistance and better define patients’ prognosis. We have learned from published trials the importance of obtaining the tumor molecular portrait at baseline to not lose patients due to their clinical deterioration. Moreover, in order to efficiently use precision medicine in oncology, we have learned from previous trials the importance of defining, among the alterations identified, which are those drivers for tumor growth, and to inhibit with specific anticancer drugs. The design of trials evaluating different targets in different tumors has allowed testing new drugs and their preliminary efficacy, thus sparing costs and reducing time. Finally, the creation of consortia has been a successful strategy to detect molecular alterations at a low prevalence.

When the accrual from all the ongoing trials are completed, it is expected the development of a global database, including the outcomes on personalized medicines, the identification of large and significant differences in small molecularly selected groups of patients, and eventually the accelerated approval of new drugs. Only creating national datasets might help to favor the discussion with the regulatory authorities for reimbursement strategies.

## Figures and Tables

**Figure 1 diagnostics-10-01092-f001:**
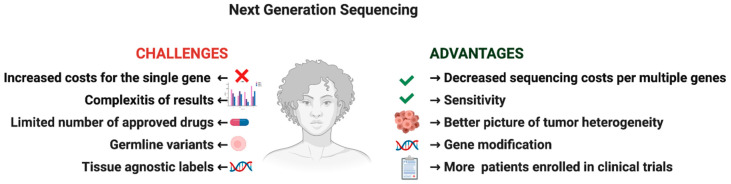
Next-generation sequencing: Advantages and pitfalls.

**Figure 2 diagnostics-10-01092-f002:**
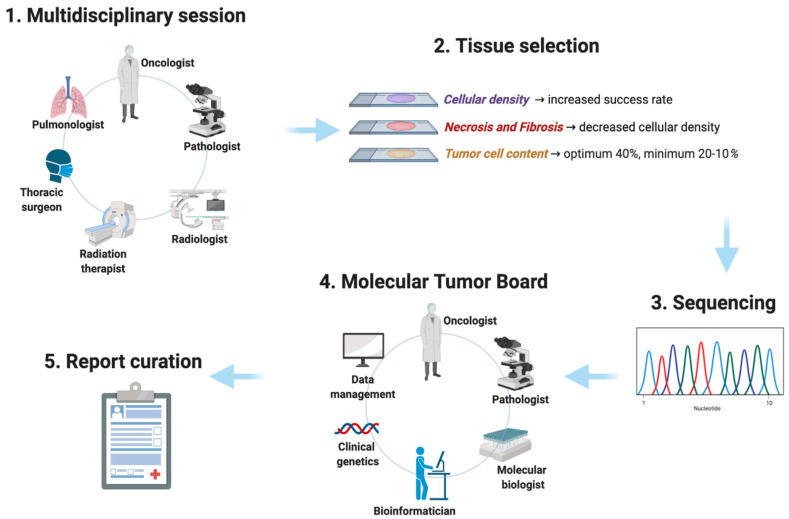
Workflow from sample selection to next-generation sequencing (NGS) analysis and genomic report.

**Table 1 diagnostics-10-01092-t001:** Level of actionability of the molecular variants.

TIER 1 Variants of Strong Clinical Significance	TIER 2 Variants of Potential Clinical Significance	TIER 3 Variants of Unknown Clinical Significance	TIER 4 Benign or Likely Benign Variants
Therapeutic, Prognostic, and Diagnostic Value	Therapeutic, Prognostic, and Diagnostic Value		
Level A evidence FDA approved therapy included in guidelines	Level C evidence FDA-approved therapies for different tumor types or investigational therapies Multiple small, published studies with consensus	Not observed at a significant allele frequency in the general or specific subpopulation databases or pan-cancer or tumor-specific variant databases No convincing published evidence or cancer association	Observed at significance allele frequency in the general or specific subpopulation databases No existing published evidence of cancer association
Level B evidence Well-powered studies with consensus from experts in the field	Level D evidence Preclinical trials or case reports without consensus		

**Table 2 diagnostics-10-01092-t002:** Studies exploring NGS in patients with solid tumors and non-small cell lung cancer (NSCLC).

Trial (Reference)	Patients	Biomarkers	Enrollment Drugs
BATTLE 2 [17]	200 patients with NSCLC, EGFR WT, EML4-ALK-, progressing to standard therapy 154 KRAS mutated	mRNA GeneChip Human Gene 1.0 ST Array from Affymetrix + NGS Foundation Medicine	enrollment closed Erlotinib Erlotinib + MK-2206 Selumetinib + MK-2206 Sorafenib
MOSCATO 01 [16]	1035 patients with solid tumors progressing to at least one line 170 with lung tumor	NGS	enrollment closed ALK AR Cell cycle DNA damage EGFR 2 ERBB2 FGFR IDH IGF1R KIT MAPK MDM2 MET NOTCH PI3K–AKT–mTOR
TAPUR [18,19,20]	3123 patients with advanced solid tumors, non-Hodgkin lymphoma, multiple myeloma progressing to standard therapy	molecular testing in a laboratory under the Clinical Laboratory Improvement Amendments and accreditation by the College of American Pathologists	enollment ongoing Crizotinib Palbociclib Sunitinib Temsirolimus Trastuzumab and Pertuzumab Vemurafenib and Cobimetinib Cetuximab Dasatinib Regorafenib Olaparib Pembrolizumab Nivolumab and Ipilimumab
S1400 Lung-MAP [21,22,23,24] (NCT02154490)	10,000 patients with squamous NSCLC, progressing to first line therapy	NGS (200 genes)	enrollment ongoing Docetaxel Durvalumab Erlotinib AZD4547 Ipilimumab Nivolumab Palbociclib Rilotumumab Talazoparib Taselisib Tremelimumab
SAFIR02_Lung (NCT02117167)	650 patients with NSCLC, EGFR WT, EML4-ALK-, with stable disease or partial response following 4 cycles of platinum based chemotherapy	DNA microarrays and NGS (50 geni, ampliSeq, ion torrent)	enrollment ongoing AZD2014 AZD4547 AZD5363 AZD8931 Selumetinib Vandetanib Pemetrexed Durvalumab savolitinib Olaparib
National Lung Matrix (NCT02664935) [25]	5467 patients with NSCLC progressing to standard therapy	NGS (143 genes)	enrollment ongoing AZD4547 Vistusertib Palbociclib Crizotinib Selumetinib Docetaxel AZD5363 Osimertinib Durvalumab Sitravatinib
NCI-MATCH (NCT02465060)	6452 advanced solid tumors, lymphomas, or multiple myeloma progressing to standard treatment	NGS (143 genes)	enrollment ongoing Adavosertib Afatinib Binimetinib Capivasertib Copanlisib Crizotinib Dabrafenib Dasatinib Defactinib Erdafitinib AZD4547 Ipatasertib Larotrectinib Nivolumab Osimertinib Palbociclib Pertuzumab GSK2636771 Sapanisertib Sunitinib Malate Taselisib Trametinib Trastuzumab Trastuzumab Emtansine Ulixertinib Vismodegib
ACC Lung	1000 patients with naive advanced NSCLC	NGS (182 genes)	enrollment ongoing Afatinib Erlotinib Gefitinib Osimertinib Alectinib Crizotinib Dabrafenib + Trametinib Pembrolizumab Atezolizumab Nivolumab Cisplatin + Pemetrexed + Pembrolizumab Carboplatin + Pemetrexed + Pembrolizumab Cisplatin + gemcitabine

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
