# Peer review of "Next Generation Sequencing in Non-Small Cell Lung Cancer: Pitfalls and Opportunities"

_diagnostics, 2020, doi:10.3390/diagnostics10121092_

Round 1
Reviewer 1 Report
This review about next generation sequencing (NGS) in non-small cell carcinomas is interesting.
There are just some remarks:
1) In this article, the interest of NGS is mainly considered for clinical trials. The value of this technique is not assessed in routine practice where most patients are not enrolled in phase II protocols. A short discussion about more simple techniques for analyses of routinely targetable mutations and biomarkers (RT-PCR for EGFR, immuno-histochemistry for ALK and PDL1) should be added. The fact that NGC on large panels of genes could be only a complementary technique in cases negative for common markers, could be discussed.
2) NGS provides many data that are useless (tiers 3 -4) and only some results can lead to an enrolment in clinical trials. The authors do not ask if, in routine practice, only “tiers 1” genomic alterations should be looked for, and if analyses of other alterations should only be for research and financed by research. Whatever the opinion of the authors is, the question must be assessed.
3) The authors address the problem of the cost and reimbursement of the treatments. They do not address the cost of the NGS itself, the cost of the tumor boards, the time spent by overworked pathologists… The authors should insist on the fact that there is a real public-health question.
4) Figure 1 is unclear. Most of items that are shown are very debatable and could be either advantage or pitfall. The cost of the technique is not mentioned and is probably still a pitfall as discussed in my previous remarks. “Reduced time for analysis” can be considered true if one takes into account the number of data provided. However, PCR for EGFR mutations and immunohistochemistry for ALK translocation are much faster than NGS that is not performed in all centers.
5) Lastly some abbreviations should be avoided: page 2: “SNV and indels”; page 3: “MTBs”
Author Response
Dear reviewer,
thank you for your comments and suggestions, useful to improve the manuscript.
As you suggested in point 1, we have included in the paragraph "Comments and conclusion" a short paragraph about the differences between NGS and standard methods. In the same paragraph we have also made a comment on the tiring and the costs.
We have modified figure 1 as you suggested and deleted the abbreviations.
Reviewer 2 Report
In this review by Lazarri et al., the authors have submitted a a comprehensive review on NGS in NSCLC: pitfalls and opportunities, that is well written and easy to understand.
Overall, the manuscript is thorough and well written. However, the manuscript can benefit from an expansion of the future directions under the conclusions section, suggesting the scope for NGS in NSCLC in the future and the application of NGS in oncology.
Author Response
Dear reviewer,
thank you for your comments and suggestions to improve the manuscript.
In the paragraph Comments and conclusions we have introduced the future perspectives of NGS.